# Single Nucleotide Polymorphism Discovery and Genetic Differentiation Analysis of Geese Bred in Poland, Using Genotyping-by-Sequencing (GBS)

**DOI:** 10.3390/genes12071074

**Published:** 2021-07-14

**Authors:** Joanna Grzegorczyk, Artur Gurgul, Maria Oczkowicz, Tomasz Szmatoła, Agnieszka Fornal, Monika Bugno-Poniewierska

**Affiliations:** 1Department of Molecular Biology of Animals, National Research Institute of Animal Production, Balice n., 32-083 Kraków, Poland; joanna.grzegorczyk@iz.edu.pl (J.G.); tomasz.szmatola@iz.edu.pl (T.S.); agnieszka.fornal@iz.edu.pl (A.F.); 2Center for Experimental and Innovative Medicine, University of Agriculture in Kraków, Al. Mickiewicza 24-28, 30-059 Kraków, Poland; artur.gurgul@urk.edu.pl; 3Department of Animal Reproduction, Faculty Anatomy and Genomics of Animal Breeding and Biology, Agricultural University in Cracow, Al. Mickiewicza 24-28, 30-059 Kraków, Poland; monika.bugno-poniewierska@urk.edu.pl

**Keywords:** goose, *Anser*, White Kołuda^®^ goose, conservative flocks, genotyping-by-sequencing (GBS), SNP discovery

## Abstract

Poland is the largest European producer of goose, while goose breeding has become an essential and still increasing branch of the poultry industry. The most frequently bred goose is the White Kołuda^®^ breed, constituting 95% of the country’s population, whereas geese of regional varieties are bred in smaller, conservation flocks. However, a goose’s genetic diversity is inaccurately explored, mainly because the advantages of the most commonly used tools are strongly limited in non-model organisms. One of the most accurate used markers for population genetics is single nucleotide polymorphisms (SNP). A highly efficient strategy for genome-wide SNP detection is genotyping-by-sequencing (GBS), which has been already widely applied in many organisms. This study attempts to use GBS in 12 conservative goose breeds and the White Kołuda^®^ breed maintained in Poland. The GBS method allowed for the detection of 3833 common raw SNPs. Nevertheless, after filtering for read depth and alleles characters, we obtained the final markers panel used for a differentiation analysis that comprised 791 SNPs. These variants were located within 11 different genes, and one of the most diversified variants was associated with the *EDAR* gene, which is especially interesting as it participates in the plumage development, which plays a crucial role in goose breeding.

## 1. Introduction

Goose breeding in Poland is an essential, increasing branch of poultry production. The Polish goose constitutes economically significant livestock that is distributed worldwide and, above all, it is well-known for its excellent properties of meat and plumage [1,2,3]. In Poland, most geese belong to 14 breeds, which are included in the national conservation program of animal genetic resources. The program constitutes a reservoir of valuable genes. The collected breeding material could be useful for evolutionary research and biodiversity improvement programs. In Poland, one of the priorities in goose breeding is the preservation of the conservative flocks’ genetic diversity and their unique, locally developed genetic traits, fitted to local economic and climate conditions [2]. There are 14 goose breeds (Kielecka (Ki), Podkarpacka (Pd), Garbonosa (Ga), Pomorska (Po), Rypinska (Ry), Landes (La), Lubelska (Lu), Suwalska (Su), Kartuska (Ka), Romanska (Ro), Slowacka (Sl), Kubanska (Ku), Zatorska (ZD-1) and Bilgorajska (Bi)) included in the national program for the protection of animal genetic resources and they are named after the region of their origin. Importantly, the breeds Kubanska, Slowacka, Landes and Romanska originally came from foreign countries, Russia, Slovakia, France and Denmark, respectively, but they were for several decades bred in Poland and adopted to local conditions. Customarily, geese breeds could be also divided according to their morphological traits, such as body weight (heavy and light geese) or plumage color (uncolored or spotted) [4]. However, one of the most significant commercial breeds is the White Kołuda^®^ goose; originally bred in The National Research Institute of Animals Production, it represents 90% of the goose population in Poland [1]. It was selected to enhance meatiness and reproduction traits, but also to improve resistance to diseases. The Polish goose population is an exciting material for population genetics and evolution studies as Poland is the largest producer of goose meat, down and feathers in Europe [1,3]. Moreover, it constitutes a poorly explored gene pool, within which there could be molecular markers useful in identifying geese products.

Commonly used markers for population genetics are single nucleotide polymorphisms (SNPs). SNPs are single base changes in a DNA sequence and are widely applied for genetic studies because of their biallelic character, abundance and dense distribution across the genome, and low mutation rate [5]. SNPs can be located in non-coding and coding DNA sequences, leading to codons substitution, stopping codons occurring or causing splicing sites disturbance. Before an SNP can be considered a molecular marker, it requires knowledge of its character and surrounding DNA sequence. For SNP discovery, mainly whole genome or enrichment-based sequencing techniques are being used. Microarrays are the technology determining the genotypes of millions of SNPs in parallel, which were beforehand identified and validated using high throughput sequencing technologies [6]. However, because of the high production costs, microarrays are designed mainly for advanced experiments involving model species, which can be commercially efficient. There is a limitation in the use of microarrays for non-model species, such as goose, due to the lack of specific DNA chip platforms for genotyping. Most importantly, the goose genome is poorly developed; therefore, microarrays manufacturing would be difficult to achieve in this case. What is more, design of the custom array would be economically ineffective for small studies in which only small population samples are being analyzed. An alternative to microarrays could be genotyping-by-sequencing (GBS) as a strategy for genome-wide SNP discovery and population-wide genotyping. The GBS is also applicable in non-model species because there is no need to locate the identified SNPs in the reference genome sequence. The GBS is technically uncomplicated, and with high multiplex abilities, it can be applied in any species with a low per-sample cost. The technique is based on next-generation sequencing (NGS) of fragment subsets created by specific restriction enzymes (REs). Restriction enzymes are responsible for reducing the genome complexity by non-random cutting in different fractions of the genome. As a result, only fragments of the genome are sequenced, which are associated with enzyme cut sites located not far apart in genome sequence (up to 1000 bp). This allows for achieving higher multiplexing of samples for sequencing because it reduces genome complexity and targeted region sizes. In addition, the GBS library preparation is quite simple, fast and highly reproducible [7]. From the practical point of view, the GBS method is based only on a few laboratory processes. Firstly, the random digestion of genomic DNA using frequently cut restriction enzyme is performed, as well as the ligation of the beforehand designed specific adaptors (indexed, containing unique barcodes) to the sticky ends of received fragments. In the next steps, the ligation products are amplified, purified, multiplexed and finally sequenced using a high-throughput next-generation sequencer [7,8].

Currently, genotyping-by-sequencing is an often-used method for small projects or initial research worldwide. However, the GBS method has some potential drawbacks, such as uneven genome coverage and a high percentage of the intergenic variants. Frequently, low coverage sequencing applied in some experiments leads to a high amount of missing data or genotyping errors, resulting in false genotypes. Furthermore, it is extremely important to choose the appropriate restriction enzyme for a given species. Additionally, the reduction in genome complexity using restriction enzymes means that, in case of any mutation at the restriction site, the genomic DNA of this region is not available to be PCR amplified and, consequently, the SNPs of this region will become undiscovered [9,10].

In the beginning, GBS was mainly applied to plants, but it has also found application in SNP detection and genotyping in animal genomics. The GBS has been successfully used in chicken, pig, cattle, sheep and horse [8,11,12,13,14,15,16,17,18,19,20]. Moreover, it is important to mention that GBS was also successfully used in Pekin duck, which is genetically close to a goose as they are in the same biological family, Anatidae [21]. Gurgul et al. (2019) also indicated a satisfying result of GBS being used in woodpeckers [22].

The present work tests a cost-effective genotyping procedure based on low coverage NGS on goose breeds maintained in Poland for single nucleotide polymorphism discovery and genetic differentiation analysis. Our research is the first attempt in the application of genotyping-by-sequencing to geese species.

## 2. Materials and Methods

### 2.1. Material

The experimental material originates from the National Research Institute of Animal Production, Department of Water Fowl Breeding in Kołuda Wielka and Dworzyska. We collected 240 plumage samples from 12 geese breeds: Kielecka (Ki), Podkarpacka (Pd), Garbonosa (Ga), Pomorska (Po), Rypinska (Ry), Landes (La), Lubelska (Lu), Suwalska (Su), Kartuska (Ka), Romanska (Ro), Slowacka (Sl) and Kubanska (Ku)—20 plumage samples from each breed. We collected 4 plumage samples from White Kołuda^®^ geese, and additionally, we sourced 96 blood samples from White Kołuda^®^ geese. In total, 340 samples were taken for analysis. The samples were selected randomly from different cages and flocks. Moreover, the samples were taken from male and female individuals. The examined blood samples were collected during the routine slaughter of geese in Kołuda Wielka, so the ethical approval was not needed for this research.

### 2.2. DNA Extraction, GBS Library Preparation and Sequencing

DNA was extracted using the Sherlock AX Isolation KIT (A&A Biotechnology, Gdynia, Poland) according to the provided protocol for both kinds of biological material. The calamus was taken as a source of DNA from plumage.

DNA was pooled in groups according to the breed, with ten individuals per pool. As a result, we obtained 34 pools with a final DNA concentration of 100 ng/µL (Table 1).

Prepared DNA pools were digested overnight in 37 °C with the restriction enzyme PstI-HF (100,000 U/mL, New England Biolabs, Ipswich, MA, USA). Afterwards, products of digestion were ligated with 4.8 ng of adaptors using T4 DNA ligase (400 U/µL, New England Biolabs, Ipswich, MA, USA) in 60 min incubation in 22 °C. In our analysis, we used 48 indexed adapters designed with GBS Barcode Generator software as described in the previous study [8]. The ligation mixture was purified using QIAquick PCR Purification Kit (Qiagen, Hilden, Germany) according to the standard protocol. Purified libraries were finally amplified in PCR reaction with universal primers (12.5 pmol/µL) and Taq Master Mix (New England Biolabs, Ipswich, MA, USA). After another round of purification, obtained libraries were qualitatively examined using Tape Station 2200 system (Agilent Technologies, Santa Clara, CA, USA) and quantified by Qubit DS DNA assay (Thermo Fisher Scientific, Waltham, MA, USA). The final step of the GBS libraries preparation was normalization to 10 nM concentration and frozen in −20 °C for further use.

The obtained libraries were sequenced in a single 50 bp run on the HiScanSQ system (Illumina, San Diego, CA, USA) with the use of TruSeq v3 chemistry and v3HiSeqflowcell.

### 2.3. SNPs Detection

Raw reads obtained after sequencing were checked for quality using FastQC software. Then, the TASSEL 5 GBS v2 Discovery Pipeline [23] was performed using *Anser cygnoides* (Swan goose) genome assembly (GooseV1.0) as a reference. TASSEL filtering steps were first applied to the reads, which did not allow base quality lower than Q20 and kept reads from having a barcode and a cut site, and no N’s in the useful part of the sequence. The procedure also trimmed off the barcodes and truncated sequences that (1) had a second cut site or (2) read into the common adapter. Default TASSEL pipeline parameters were used for SNPs detection, except kmer Length which was set to 40 and had a minimum count of reads for a tag with an output set to 5. Tags were mapped to the reference genome using a Bowtie2 aligner [24].

Polymorphisms resulting from the default TASSEL Production Pipeline were initially filtered to remove indels and multiallelic markers with an average coverage lower than 170 reads (5 per pool on average). Additionally, individual pool ‘genotypes’ were removed if the coverage was lower than 5 reads. Then, SNPs located on X and Y chromosomes were excluded because of the uneven distribution of sex within pools.

### 2.4. Genetic Differentiation Analysis and SNPs Annotation

The read counts obtained for separate alleles were used to calculate allele frequencies for individual SNPs in separate pools. Then, overall allele frequencies were calculated for breeds as an average for all pools sequenced per breed. In rare cases of no coverage for some SNPs in single pools, allele frequencies were derived from the remaining pools.

Frequency data were used to perform breeds differentiation analysis by principal components analysis (PCA) and hierarchical clustering in ClustVis software [25]. Then, Nei’s genetic distances [26] between all breeds were calculated from allele frequencies using Gendist software [27] implemented in the Phylip package [28]. The distances were calculated using a reduced SNPs set with the exclusion of 338 markers with missing data for some breeds (which are not allowed in the Gendist software). The obtained genetic distances were used to create a cladogram based on the neighbour-joining method [29]. Finally, SNPs were annotated using the Variant Effect Predictor tool (ENSEMBL) with respect to the GooseV1.0 genome assembly.

To identify variants that were mostly associated with the breeds differentiation, the variance of allele frequency was calculated for each SNP. The 10% of SNPs with the highest variance among breeds were further screened for colocalization with genes. The associated genes were analyzed for their functions using the Panther Classification System [30] and human orthologs. An over-representation test in GO categories was performed with respect to all known human genes using the Fisher’s exact test.

### 2.5. SNP Validation

GBS accuracy was established by EDAR gene sequencing in 96 selected individuals from each goose breed examined in this analysis. Primers were designed using the Primer 3 software, based on the DNA sequence for EDAR of *Anser cygnoides* (F′AAGCACTGGCCTTCTCACAA; R′AGAGCGGAGCCTAGACTGAA). The expected fragment was amplified using HotStarTaq DNA Polymerase (1000 U) (Qiagen, Hilden, Germany) according to the attached protocol, then the one-step enzymatic purification of the PCR product was performed using ExoBap (Eurx, Gdańsk, Poland) and sequenced using BigDye^®^ Terminator v3.1 Cycle Sequencing Kit (Thermo Fisher Scientific, Waltham, MA, USA). DNA sequencing products were separated using the 3500xl Genetic Analyzer (Applied Biosystems, Foster City, CA, USA) and analyzed in BioEdit software.

## 3. Results

### 3.1. Reads and SNPs Statistics

The applied sequencing depth allowed generating from 14,545 to 1,157,023 raw reads per pool. After initial filtering and TASSEL preparatory step, from 22,561 to 1,493,738 good, barcoded reads (with a *PstI* cut site detected) were used for tags detection and SNPs discovery (Appendix A).

The initial production pipeline allowed for the detection of 3833 SNPs. Further filtering for read depth and alleles characters (multiallelic, indels) removed 3042 SNPs. The final markers panel used for differentiation analysis comprised of 791 high-quality SNPs. The average coverage of the filtered SNPs was 840.7 and ranged from 171 to 25,962.

The analysis of the SNP parameters showed that their expected heterozygosity ranged from 0.02 to 0.499 and was 0.215 on average. The evaluation of missing genotypes showed that individual SNPs were characterized by 0% to 53% of missing genotypes, with an average of 8.1%. The list of variants used in this study is presented in Appendix A.

### 3.2. Goose Breeds Genetic Differentiation Based on GBS Data

The expected heterozygosity calculated per breed ranged from 0.16 in Ku to 0.23 in BK (Appendix A). The distribution of the major allele frequency was similar for separate breeds, and the major allele had a frequency of 0.815 on average (Appendix A). The studied geese breed genetic differentiation analysis was initially performed using the PCA method, which described about 29% of their genetic variation. This analysis showed three visible clusters of breeds. A clearly distant, genetic profile was observed for Garbonosa and Kubanska geese. Two remaining, much more closely positioned clusters, involved: (i) Pomorska, Romanska, Suwalska and Slowacka breeds, and (ii) the remaining studied breeds (Figure 1).

Similar results were obtained when the cladogram was created based on Nei’s pairwise genetic distances. This analysis showed, however, that the Romanska breed was rather more similar to the breeds accounted to the third cluster identified with PCA, including, inter alia, the White Kołuda^®^ breed. In contrary, the results obtained with hierarchical clustering were rather more similar to those obtained with PCA (Figure 2).

### 3.3. SNPs Annotation

Annotation analysis of SNPs, with respect to the genes from the GooseV1.0 assembly, showed that most of the identified variants were found in the intergenic regions (51%). A relatively high number of variants were also found in the gene introns (14.6%), as well as in the upstream gene regions, including promotes (13.3%). Among SNPs located in coding sequences, 58% were synonymous (Figure 3).

Detected variants were located within 11 different genes. The genes enriched significantly on a pointwise level and several biological processes, including ones connected with, e.g., lipid catabolic process, pancreatic juice secretion, steroid catabolic process (*CEL*), salivary gland morphogenesis (*EDAR*), pigmentation (*KIF13A*), skin epidermis development and hair cycle (*EDAR*) (Appendix A).

### 3.4. Identification of SNPs the Most Diversified among Analyzed Breeds

Based on variance distribution, 79 SNPs with the largest differences in allele frequencies between the breeds were selected for functional analysis (Appendix A). The most diversified variants were associated with only four genes, namely Tetraspanin-18 (*TSPAN18),* Kinesin-like protein KIF13A *(KIF13A),* RING finger protein 223 *(RNF223)* and Ectodysplasin A receptor (*EDAR)*, and were located within the gene introns. The most polymorphic gene was KIF13A, where 10 potential SNPs were found. The genes are connected with various biological processes, including regulation, cellular processes, metabolism, response to stimulus and signaling.

### 3.5. Validation of GBS Results by Sanger Sequencing in the EDAR Gene

Using the Sanger sequencing, we confirmed three SNPs identified by GBS and four additional SNPs located in the *EDAR* gene (Figure 4). Five of the identified SNPs were distributed widely, in almost every individual, while two of the variants (T/A; C/T) were present only in Pomorska and Kartuska, respectively. Apart from this, we have not observed any breed’s dependences between examined animals or any private SNP, representing only one breed. However, further investigation covering a larger number of samples and breeds is needed to fully explore the evolution of the *EDAR* gene in goose breeds.

## 4. Discussion

The exceptional properties of meat and feathers sourced from goose are well-known worldwide, contributing to this representative of poultry as an economically important breeding animal. Currently, goose breeding advantages are even more appreciated, making it still a growing branch of poultry production [1,2,3]. Nevertheless, the genetic diversity of goose is little explored. Recently, we performed an analysis of the goose populations maintained in Poland using microsatellite markers. The results showed that geese bred in Poland do not form separate populations as shown by genetic markers and are characterized by a high admixture level at the STR loci [31]. Furthermore, it was impossible to discriminate the White Kołuda^®^ breed from the other breeds kept in Poland, based on the STR analysis. Therefore, we decided to utilize the GBS approach to gain more power in our genetic analysis and improve upon our knowledge about goose differentiation at the molecular level.

Genotyping-by-sequencing is a genetic screening method for SNPs discovering and performing genotyping. Sequence-based genotyping provides a lower-cost alternative to microarrays for studying genetic variation and is defined as a high throughput molecular tool applied with the engagement of reasonable financial resources [7]. Previous studies have already verified the GBS method as a useful tool for bird’s population genomics [17,21]. Our research used the GBS method because of the absence of a reference genome for the goose and other high throughput genotyping tools for geese. Our target was to detect SNPs that allowed for analyzing genetic diversity in geese maintained in Poland. Moreover, according to our best knowledge, it is the first attempt to use GBS for population genomics in goose.

The markers panel used for our analysis constituted 791 SNPs, obtained from 3833 SNPs after filtering for read depth and alleles characters. It should be noted that our research is the first GBS analysis on representatives of the goose breeds. Our analysis allowed for identifying a relatively low amount of SNP compared to other GBS studies, e.g., in duck (169,209) [21]. This may be due to several reasons, for example, the selection of a single digest protocol with *PstI* enzyme, which may be not suitable for the geese genome. Nevertheless, it was previously successfully used in chicken GBS analysis, which suggests that this reason is rather unlikely [17]. Moreover, the geese genome is not fully completed and only available as scaffolds, which may lower the number of identified SNPs. The most important reason for a low number of SNPs detected in this study is the applied low coverage sequencing. This allowed detection of a lower number of SNPs than in experiments involving high coverage sequencing, however, presumably with enough power to differentiate the analyzed breeds.

The GBS analysis of the conserved flocks and the White Kołuda^®^ geese maintained in Poland allowed us to characterize their genetic diversity and mutual phylogenetic relationships. The dendrograms of examined geese showed a clear division into three distinct groups (Figure 1 and Figure 2) Garbonosa, Kubanska, and the third group, which contained all other populations. This division is in agreement with the phylogenetic origin of studied populations: Garbonosa and Kubanska geese are derived from a different ancestor (*Ansercy gnoides*) than all the other populations (*Anser anser*). Using microsatellite markers for phylogenetic analysis of the same material did not allow for such clear identification of descendants of separate species [31], which proves the advantage of the GBS method over the use of microsatellite sequences for phylogenic analysis. Our results also showed unexpected relationships between some geese breeds. Surprisingly, Pomorska and Suwalska geese, which come from Poland’s northern regions, were combined in single taxa with a Slowacka goose of a different geographical origin. Romanska goose similarly to Landes are foreign goose breeds, however, they were not clearly distinct from Polish breeds. This could indicate an admixture of those breeds with Polish geese: Lubelska and Rypinska (Figure 2A). Furthermore, Podkarpacka and Kielecka, Kartuska and White Kołuda^®^ geese were placed in the same cluster despite Podkarpacka and Kielecka come from the southern regions of Poland, while Kartuska and White Kołuda^®^ geese come from the northern regions. In conclusion, we may say that our GBS analysis agrees with the phylogenetic origin of geese, but it does not reflect their geographic origins provided by historical data [31].

Current GBS analysis and our previous studies based on microsatellite markers suggest that geese maintained in Poland are closely related to each other and genetic variation between breeds is low, making it difficult to distinguish them from each other. The present analysis shows no clear division depending on phenotypic traits such as geese body weight or plumage color. Among breeds analyzed by us, some are generally considered as heavy weight (Pomorska Kartuska, Rypinska, Suwalska, Romanska) and light weight geese (Kubanska, Garbonosa, Lubelska, Kielecka, Podkarpacka). Slowacka goose body weight is considered as medium when compared to the others. Another morphological trait that could shape genetic patterns of variation in Polish geese may be plumage color and color patterns. Pomorska, Slowacka and Romanska are characterized with white plumage while Kartuska, Podkarpacka and Rypinska are white and speckled grey or brown [29]. Contrary to expectation, these morphological features were weakly reflected in our analysis.

One of the genes associated with the most diversified genome region detected in this study was *EDAR* which had SNP variants located within the intron. This seems to be especially interesting, as *EDAR* encodes an ectodysplasin A receptor that plays a crucial role in developing ectodermal tissues, such as skin or plumage [32]. In some cases, feathers constitute a valuable export product in terms of the excellent properties used in the production of high-quality pillows, quilts and clothing appreciated all over the world [1]. Plumage and feather rudiments are formed during embryonic development in ordered patterns, particularly in flighted birds such as geese, and the EDA/ectodysplasin A receptor (*EDAR*) plays a critical role in this. The EDA signal co-operates with spreading a wave of cellular density to spark the formation of regularly spaced cell aggregation, each of which serves as the precursor to a feather [32]. We could presume that some SNPs in this gene may affect the plumage formation in goose and, in turn, it would influence the quality of the feathers. Undoubtedly, our analysis proved that the *EDAR* gene contains several single nucleotide polymorphisms which would be an interesting object to examine deeper on a larger number of animals.

## 5. Conclusions

Our findings proved that the GBS method is a useful tool for population genetics, SNP discovery and genetic differentiation analysis in non-model species (without reference genome sequence) such as goose. Moreover, our results showed that GBS-derived SNPs have more power than microsatellite markers to detect the Polish geese’s genetic differentiation. Some patterns of variation have been captured that may be related to the breeds phylogenetic origin and also some of their geographic origin, body weight and plumage color. Additionally, analysis in variation across the genome allowed us to point to the gene that is potentially responsible for the feather’s development.

## Figures and Tables

**Figure 1 genes-12-01074-f001:**
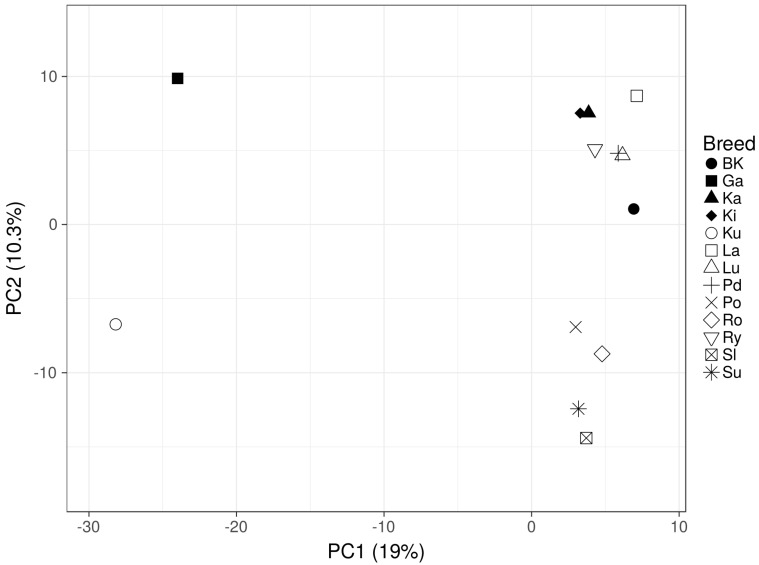
Principal component analysis based on alleles frequency for all studied breeds (White Kołuda^®^ (BK), Kielecka (Ki), Podkarpacka (Pd), Garbonosa (Ga), Pomorska (Po), Rypinska (Ry), Landes (La), Lubelska (Lu), Suwalska (Su), Kartuska (Ka), Romanska (Ro), Slowacka (Sl) and Kubanska (Ku)).

**Figure 2 genes-12-01074-f002:**
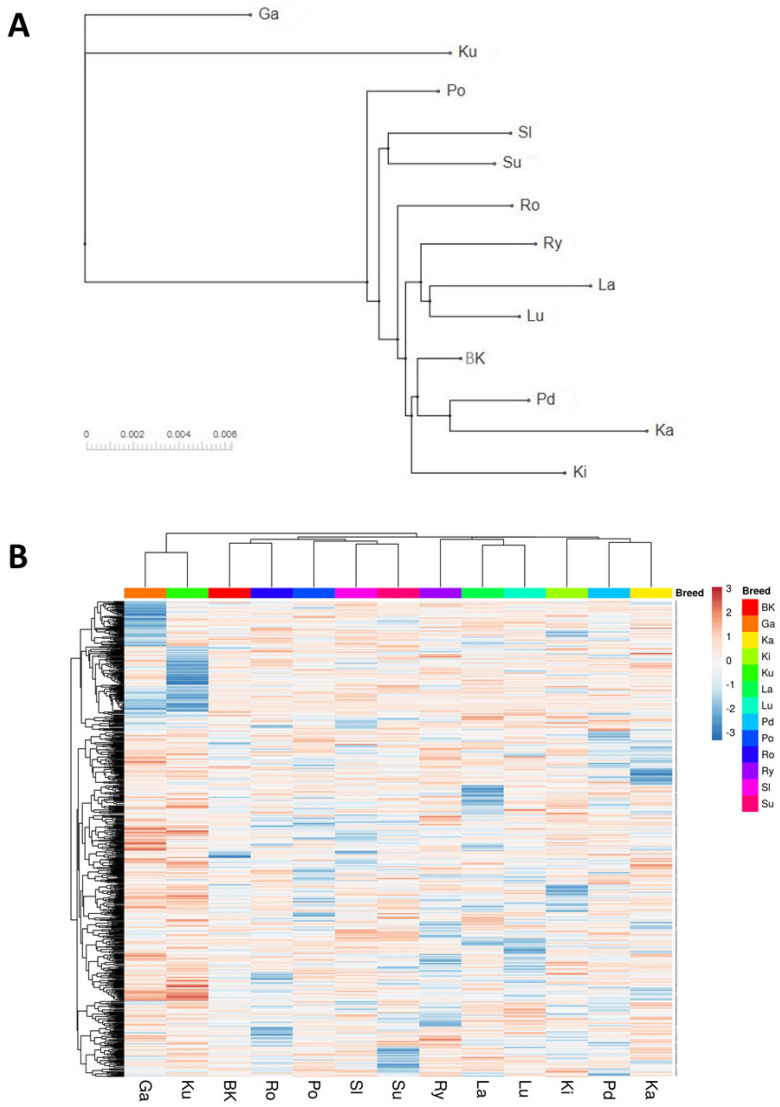
Cladogram created based on Nei’s genetic distances (**A**) and a frequency heatmap along with the hierarchical clustering of breeds and SNPs (**B**) (White Kołuda^®^ (BK), Kielecka (Ki), Podkarpacka (Pd), Garbonosa (Ga), Pomorska (Po), Rypinska (Ry), Landes (La), Lubelska (Lu), Suwalska (Su), Kartuska (Ka), Romanska (Ro), Slowacka (Sl) and Kubanska (Ku)).

**Figure 3 genes-12-01074-f003:**
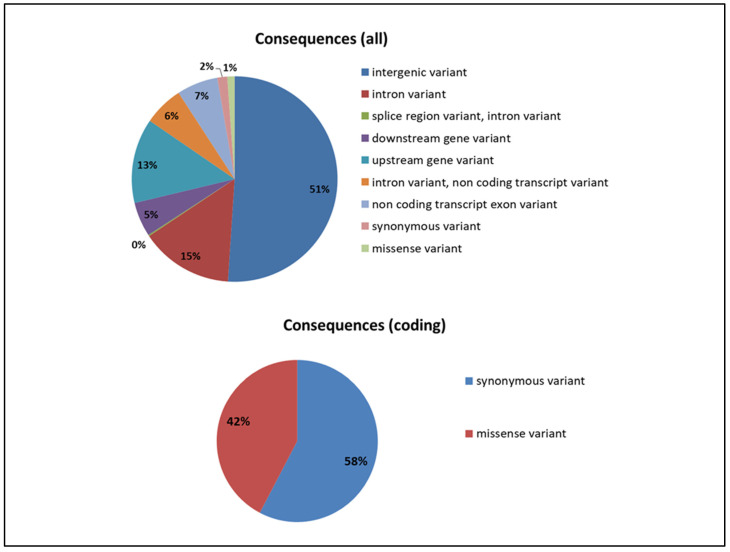
Distribution of SNPs in different functional parts of genes and genome.

**Figure 4 genes-12-01074-f004:**
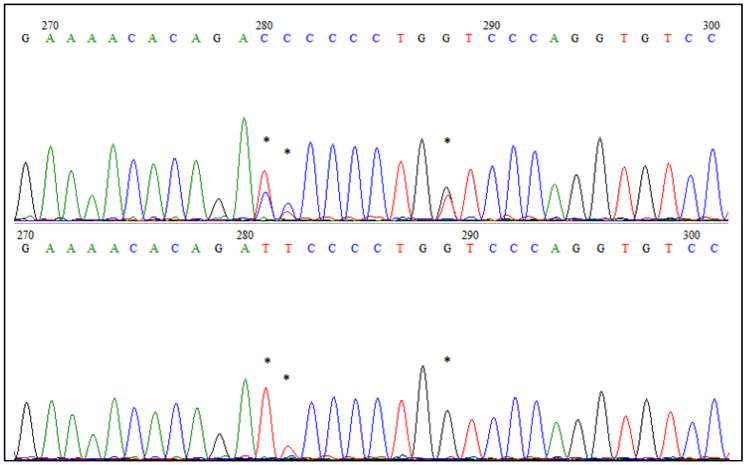
Chromatograms of fragments of the *EDAR* gene sequence after the Sanger sequencing. Asterisks indicate polymorphic SNPs, also identified by the GBS method.

**Table 1 genes-12-01074-t001:** DNA pools prepared for the experiment.

Breed	Number of Pools	Number of Individuals in Pools	Number of Individuals
White Kołuda^®^ (BK)	10	10	100
Kielecka (Ki)	2	10	20
Podkarpacka (Pd)	2	10	20
Garbonosa (Ga)	2	10	20
Pomorska (Po)	2	10	20
Rypinska (Ry)	2	10	20
Landes (La)	2	10	20
Lubelska (Lu)	2	10	20
Suwalska (Su)	2	10	20
Kartuska (Ka)	2	10	20
Romanska (Ro)	2	10	20
Slowacka (Sl)	2	10	20
Kubanska (Ku)	2	10	20
Total	34	-	340

## Data Availability

Data is contained within the article or supplementary material in “Single Nucleotide Polymorphism Discovery and Genetic Differentiation Analysis of Geese Bred in Poland, Using Genotyping-by-Sequencing (GBS)”.

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
