# Peer review of "Single Nucleotide Polymorphism Discovery and Genetic Differentiation Analysis of Geese Bred in Poland, Using Genotyping-by-Sequencing (GBS)"

_genes, 2021, doi:10.3390/genes12071074_

Round 1

Reviewer 1 Report

The research presents novel and original approach to investigate genomes of Polish goose breeds. However, there are some points which should be addressed.

The major limitation of this paper is presentation of the results. The PCA plot should be improved to make it more representing and descriptive for publication (for example, different color for each breed, legend with full names of the breeds).

To study more deeply genetic links and admixture between the breeds I recommend calculating Fst values, constructing Neighbor Net graph and performing Admixture analysis (or using Structure software to investigate population structure). Applying of these methods will increase the scientific soundness and the interest for potential readers as well as will be used for better comparison with previous results based on using STR-markers (Warzecha, J. et al 2019, https://doi.org/10.3390/ani9100737 ).

Conclusions can be improved too. P10L213-313 «Some patterns of variation have been captured that may be related to the breeds phylogenetic origin and some of geographic origin, body weight and plumage colour». This conclusion is not supported by results or discussion because no information on body weight and plumage colour is given in the paper.

P10L316 «Analysis of variation across the genome allowed additionally to point gene that is potentially responsible for feathers development and is highly diversified among Polish geese breeds». It sounds a bit contradictory and ambiguous to the results (P7L221-222) which indicated that diversified SNP variants (T/A; C/T) within the EDAR gene were found only in two breeds (Pomorska and Kartuska). Besides this finding is not discussed.

Introduction includes a detailed description of GBS methodology which is standard. The authors highlighted only advantages of the GBS for non-model species. The mentioning of some limitations of the GBS (uneven genome coverage, high percentage of the intergenic variants, etc.) will provide more interesting background for potential readers which are not familiar with the GBS technology.

In addition, I recommend adding some information on the origin, productive types or other aspects of the local goose breeds which are not widely known outside Poland.

P 7 L207 – 211 The genes corresponding to the function might be mentioned.

Reviewer 2 Report

The work entitled: "Single Nucleotide Polymorphism discovery and genetic differentiation analysis of geese bred in Poland, using Genotyping-by-Sequencing (GBS)" aims to apply genotyping by sequencing for the detection of markers useful for the discrimination of geese breeds. The study is of interest to the scientific community but needs some major revisions before being published. The introduction explains in detail the use of genotyping by sequencing approach for the detection of SNPs in non-model organisms. The authors should report an example of the cost difference between the genotyping by sequencing approach and the use of commercial SNPs arrays chips.

The materials and methods section is well written and detailed. I suggest the authors to add an analysis with the admixture software (or similar) using common SNPs between breeds. Results from admixture analysis can be useful to better discuss results at lines 274-282.

In the results section the author reported the range of the raw reads obtained from sequencing (from 14,545 to 1,157,023). The range is very wide, and the low number of reads for a given breed leads to a reduced number of SNPs to be used for the analyses. The authors must insert a table (also supplementary or in Table 1) in which to report the number of raw reads for breeds. Moreover, if the low number of reads was obtained only for one breed, I recommend eliminating it from the study to use a greater number of SNPs. At lines 171-172, 791 high-quality SNPs are reported, but in Supplementary file 1, 960 SNPs are listed. The authors should clarify this discrepancy.

Figure 1 is of poor quality and it needs to be improved. Please add colors to better identify the breeds and add a legend with name of the breeds.

Figure 2 B is of poor quality and the abbreviations of the breeds name are not legible.

In line 213, the authors state that 79 SNPs were selected based on their frequency across breeds and that they are reported in Supplementary file 1. Looking at the supplemental file it is not clear what these SNPs are. Authors should highlight these markers or report them in another sheet.

Minor comments:

In line 23 of the abstract please replace “fileting” with “filtering” and the same at line 170.

In lines 215-216 please, write the full name of the genes before the acronym.

Line 305: what do the authors mean by SNP high-density gene? Maybe the authors mean that there are several polymorphisms in the EDAR gene.

Reviewer 3 Report

The authors have presented a nice piece of work and new set of tool for identifying SNPs. However, I have certain suggestions that I would like authors should address them in the manuscript

  1. In the results authors should also mention about the level of within species polymorphisms
  2. In the materials and method authors should also mention whether samples where randomized before genotyping. Also, genotyping of both sexes were done separately or it is mixed
  3. The level of polymorphisms was calculated using polymorphisms on 3-4 genes. Why not more genes were taken identify the level of polymorphisms.
  4. What is the filtering  criteria that authors used to filter the raw reads. Please do explain in the material and methods sections.   
  5. In each species what are the genes which show large no. of SNPs. Please mention name of the genes and their level of polymorphisms.                                                                                                                                                                                                                                                                                                                                                                                                                                                                                                                                                                                                                                                                                                                                                                                                                                                                                                                                                                                                                                                                                                                                                                                      

Round 2

Reviewer 2 Report

thanks to the comments of the reviewers and the efforts of the authors, the manuscript has been greatly improved. The work is interesting in principle and informative for scientists in the field. Therefore, in my opinion, the manuscript  is acceptable for publication.

Reviewer 3 Report

No comments